# Mixing and Matching Chromosomes during Female Meiosis

**DOI:** 10.3390/cells9030696

**Published:** 2020-03-12

**Authors:** Thomas Rubin, Nicolas Macaisne, Jean-René Huynh

**Affiliations:** 1Collège de France, PSL Research University, CNRS, Inserm, Center for Interdisciplinary Research in Biology, 75005 Paris, Francenicolas.macaisne@ijm.fr (N.M.); 2Institut Jacques Monod, UMR7592, 15 rue Hélène Brion, 75013 Paris, France

**Keywords:** homologous chromosomes, pairing, synaptonemal complex, cytoskeleton, LINC

## Abstract

Meiosis is a key event in the manufacturing of an oocyte. During this process, the oocyte creates a set of unique chromosomes by recombining paternal and maternal copies of homologous chromosomes, and by eliminating one set of chromosomes to become haploid. While meiosis is conserved among sexually reproducing eukaryotes, there is a bewildering diversity of strategies among species, and sometimes within sexes of the same species, to achieve proper segregation of chromosomes. Here, we review the very first steps of meiosis in females, when the maternal and paternal copies of each homologous chromosomes have to move, find each other and pair. We explore the similarities and differences observed in *C. elegans*, *Drosophila*, zebrafish and mouse females.

## 1. Introduction

The oocyte is the final product of germ cell differentiation in females. It is an end and, at the same time, a new beginning for sexually reproducing organisms. Germ cell differentiation starts with the formation of primordial germ cells (PGCs) at embryonic stages. PGCs associate with somatic cells to form gonads, and increase their numbers by mitosis to create a pool of precursor cells [1]. Depending on the species, these precursors can either become stable germline stem cells or differentiate by undergoing a limited number of mitosis. At the end of this mitotic phase, germ cells enter meiosis. Throughout the adult life of males, all differentiating germ cells complete meiosis and become sperm. In females, depending on the species, all or only a few germ cells complete meiosis and differentiate as oocytes. The remaining germ cells become support cells for nurturing the oocyte. In mammals, such as humans and mice, these early stages of differentiation are limited to fetal ovaries, which makes their study challenging, while in *C. elegans*, *Drosophila* and zebrafish, oocytes are produced throughout adulthood.

Meiosis is a special type of cell division that is specific to germ cells, whereby diploid cells undergo two rounds of nuclear division to produce four haploid cells [2,3]. Homologous chromosomes (homologues) segregate during the first division (meiosis I) and chromatids separate during the second division (meiosis II). Each of the four daughter cells eventually inherits one set of chromatids. In order to segregate properly at meiosis I, each chromosome first needs to pair with its homologue. Pairing defines the association of homologous chromosomes. It can be initiated by a looser coupling or alignment of homologues. Pairing is then stabilised by the polymerisation of a proteinaceous scaffold called the synaptonemal complex (SC), which holds together homologous axes (synapsis), and promotes genetic recombination. Recombination is induced by the formation of developmentally programmed double-strand breaks (DSBs), which are repaired using the homologous sequence as template, resulting in reciprocal exchanges of genetic material between paternal and maternal chromosomes. Meiotic DSBs are induced by the topoisomerase-like Spo11, which is conserved in all species studied so far. Some species require DSBs to initiate homologous pairing, while it can be completely dispensable in other species, or only required for late stages of pairing (Table 1). Exchanges, or crossovers, allow for the formation of physical links, chiasmata, which maintain homologues associated in pairs upon depolymerisation of the SC. Associated homologues then orient toward opposite poles of the spindle. The initial pairing of homologues is thus crucial for correct segregation from each other at anaphase I. The pairing process, synapsis (the polymerisation of the SC), and recombination occur earlier at the onset of meiosis, during an extended prophase I (Figure 1). Five stages are distinguished during prophase I: leptotene is when homologues start to condense and become visible upon staining; zygotene is when homologous chromosomes start to synapse; pachytene marks the completion of synapsis along the full length of every chromosome pair; at diplotene, the SC depolymerises and homologues remain linked only by chiasmata; diakinesis is the final stage of prophase I, when chromosomes are condensed enough to make chiasmata detectable. 

At least three different features appear common to the pairing process in most organisms (Table 1). The first is chromosome movements [76]. Homologues are thought to be randomly positioned in the nuclear space. Thus, in order to pair, they must first move from their initial position to meet and find their homologues. Depending on the species, chromosomes can move individually, or follow general nuclear movements, or a combination of both. The second is the links with the cytoplasmic cytoskeleton [76]. Chromosome movements are driven by the cellular cytoskeleton, usually microtubules but also actin. A conserved family of transmembrane proteins (SUN/KASH, also known as LINC complex) has been found to link cytoskeletal forces to chromosomes through the two layers of the nuclear envelope (NE). Indeed, at this stage, the NE is intact, chromosomes are nuclear, while the cytoskeleton is cytoplasmic. The third is the clustering of chromosomes. Global clustering of telomeres or centromeres of all chromosomes has been observed in many species, referred to as telomeres bouquets or centromeres coupling, respectively. These non-specific interactions are hypothesized to facilitate homologues recognition and pairing. These three features have been best characterized in unicellular organisms such as the budding yeast *S. cerevisiae* and the fission yeast *S. pombe*. These two simple model systems will serve as very useful reference points to compare with more complex multicellular organisms, which we address later (Figure 1).

*Saccharomyces cerevisiae* has 16 pairs of chromosomes and clusters its telomeres as a bouquet upon entry into meiosis. Around the same time, centromeres become associated (or coupled). Both processes are independent of DSB formation and are not driven by homology. Nevertheless, it requires Zip1 at centromeres, a protein of the synaptonemal complex, which only polymerises at later stages [4]. These early associations later become homology-dependent and require DSBs to mature as stable homologous pairings. Efficient pairing necessitates sustained and rapid movements of chromosomes from leptotene to pachytene. These movements, as well as telomere clustering, depend on Ndj1p and Csm4p, which both accumulate to telomeres and whose function is to allow the anchoring of telomeres to the nuclear envelope. Ndj1p interacts with the SUN-domain protein Mps3p that links nuclear components to the cytoskeleton [36,43,44]. No clear equivalent of a KASH domain-like protein that would bridge the inner-nuclear envelope protein Mps3p to the cytoskeleton has been identified, but it could be Mps2p [44,56]. In the budding yeast, meiotic forces are ensured by actin, which localises close to the spindle pole body (SPB), tangent to and following the curvature of the nuclear envelope [37,40]. Microtubules or dynein are not involved. Movements occur through polymerisation of actin rather than via an active “sliding” of chromosome ends over actin cables [36,37]. Interestingly, in pachytene, telomeric movements are accompanied by nuclear envelope deformations [37,38]. Actin forces seem to be applied towards the nuclear envelope, causing its deformations, rather than directly on chromosome ends. Indeed, chromosome end movements and envelope deformation can be correlated as they appear with the same dynamics, but while telomere movements depend on nuclear envelope deformation, deformations are not suppressed when chromosome movements are impeded through the depletion of the telomeric anchor Ndj1p [37,77]. Chromosome ends usually move in groups, with a leading chromosome end, which can travel abruptly 0.5 µm to 1 µm at 0.3–0.5 µm per second, followed by a coordinated movement of the telomeres to which the leader is associated. These rapid movements of a telomere cluster can affect the trajectory of non-associated nearby chromosomes or groups of chromosomes [37,44]. Like in other organisms, these movements are thought to promote stable homologous interactions and to prevent unstable associations [36,37,38,44]. The difference between movements in zygotene (telomere clustering), and in pachytene (abrupt movements) are proposed to be due to differences in chromosome compaction. Zygotene chromosomes may be looser and more flexible than the stiffer, more condensed pachytene chromosomes [37].

The fission yeast, *Schizosaccharomyces pombe*, has three pairs of chromosomes and does not form a synaptonemal complex. Prior to meiotic entry, centromeric kinetochores are grouped, and linked in a Csi1-dependent manner to the SPB through the LINC complex (composed of the SUN-domain protein Sad1, and the KASH-domain proteins Kms1 and Kms2). Telomeres are also attached to the nuclear membrane but located away from the SPB (Rabl orientation) [58,59,60]. A complex composed of Taz1, Rap1 and Bqt1/2 allows for the attachment of telomeres to microtubule organising centers called telocentrosomes (consisting of γ-tubulin and dynein), via the transmembrane LINC complex [48,49,50]. Upon entry into meiosis, directly following karyogamy (fusion of the two nuclei), telomeres slide over the nuclear envelope to the SPB, from which centromeres detach, thus forming a bouquet of chromosomes. The typical stages of meiotic prophase have not been characterised in *S. pombe* and this period is referred to as the horsetail stage. Centromere detachment is dependent on telomere clustering, and both events are regulated by the kinases MAP and Pat1 [50,64]. The nucleus then elongates and goes back and forth in the cell, in a movement resembling a horsetail, through the activity of dynein over microtubules nucleated from the SPB. Horsetail movements stop after approximately 2 h, just before the first division, and the telomere cluster is resolved [6,20,78,79]. The formation of the bouquet has been shown to be essential for chromosome pairing. It gives the kinetochores the properties that are compatible with the meiotic divisions, and with proper recombination between homologous chromosomes. Moreover, as nuclear envelope breakdown does not occur during meiotic divisions in *S. pombe*, the interaction between chromosomes and the SPB through the LINC-complex is essential to allow the SPB to penetrate the nuclear envelope in order to nucleate the spindle [80,81]. With these two unicellular model systems in mind, we now explore how homologous chromosomes move in *C. elegans*, *Drosophila*, zebrafish and mouse (Figure 1).

### 1.1. Caenorhabditis Elegans

The nematode *Caenorhabditis elegans* has been at the forefront of our understanding of meiotic chromosome movements and pairing in multicellular organisms. In this worm, germline cells are produced by a pool of mitotic stem cells located at the distal tips of the gonad arms. Cells switch to the meiotic program in a region called the transition zone, where the chromatin adopts a peculiar half-moon shape while being pushed by the nucleolus towards one side of the nucleus (close to the microtubule-organising center) [21,29,30]. Meiocytes then undergo prophase I while progressing along the gonad arm. They progress to metaphase I upon fertilization while they pass through the spermatheca. *C. elegans* has six pairs of holocentric chromosomes, i.e., which centromeres are distributed along the entire chromosome length. Their pairing initiate upon entry in meiotic prophase I in the transition zone. In the nematode, alignment, pairing and synapsis of homologous chromosomes do not rely on the formation of meiotic DSBs. Homologues pair through regions of repetitive sequences located near one end of each chromosome called pairing centers (PCs), rather than through their telomeres. Chromosome-specific C2H2 zinc finger proteins bind the PCs: HIM-8 for chromosome X, ZIM-2 for chromosome V, ZIM-1 is recruited on both chromosomes II and III, and ZIM-3 for chromosomes I and IV [51,52]. Each of these proteins is able to recognize a specific 12-bp nucleic motif repeated on PCs [11]. Loss of a PC prevents the pairing as well as the synapsis of corresponding chromosomes, which can lead to their non-disjunction. Interestingly, while non-homologous chromosomes that share the same ZIM protein do not pair, the presence of homologous PCs is sufficient to ensure both pairing and synapsis of chromosomes that are not homologous along the whole length due to genetic rearrangements. This suggests that PC-bound zinc-finger proteins are not the sole determinant of chromosome identity but that they may reside inside the PC itself [12,13,14].

While PC-repeats are sufficient for the loading of PC-binding proteins, the localisation of autosomal PCs on the nuclear envelope is dependent on the activity of the kinase CHK-2. Interestingly, the localisation of the X chromosomes to the nuclear envelope is not affected by CHK-2 depletion [52]. The meiotic Polo-like kinases PLK-1 and PLK-2 are also essential players for chromosomal movements. These proteins are partially redundant and both are detected to the nuclear periphery at meiotic entry, although they do not colocalize consistently [65]. PLKs are not essential for the localisation of PCs on the nuclear envelope, but the recruitment of PLK-2 by the PC-binding proteins at the nuclear periphery allows the meiosis-specific phosphorylation of SUN-1 (on serine 12) by PLK-2 itself, and induces a relocalisation of the SUN and KASH-domain proteins to regions of the nuclear envelope where PCs are located [65,66]. The transmembrane SUN/KASH protein complex, like in other organisms, acts as a link between the chromosomes and the cytoskeleton, allowing the transfer of dynein-generated forces [46]. This results in highly dynamic, but random movements of the PCs, either in isolation or as small groups, while other chromosomal regions appear to be less mobile [30]. Although PCs can group, they never cluster to form a meiotic bouquet-like structure [42]. PC movements are thought to provoke random interactions of homologous PCs until they are stabilised by the polymerisation of the synaptonemal complex central region between homologous axes [29]. These movements are meant to prevent non-homologous interactions by shuffling unstable alignments [41,42]. Indeed, when chromosome movements are impeded (e.g., in mutants for SUN/KASH, or by preventing dynein function), pairing is greatly reduced and non-homologous synapsis is observed [41,46,82,83]. Interestingly, while in mutants for a single PC-binding protein, there is no synapsis for the corresponding pair(s) of homologous chromosomes; depletion of all four PC-proteins does not prevent the polymerisation of the synaptonemal complex. In such cases, however, synapsis occurs on folded-back univalent chromosomes [66]. This finding argued for the existence of a homolog-pairing checkpoint relying on forces transduced from the cytoskeleton to the nucleus [14].

Other proteins have been reported as essential to chromosome pairing in *C. elegans*. HAL-2/HAL-3 is an orphan protein complex located in the nucleoplasm. The loss of these proteins induces misregulation and mislocalisation of PLK-2, defective homolog pairing and abnormal loading of synaptonemal complex central region proteins onto unpaired axes. The HAL-2/HAL-3 complex is thought to be a regulator of meiotic Polo kinases PLK-1/2 [67,68]. FKB-6 is a DAF-21/Hsp90 co-chaperone, homolog to mammalian FKBP52 that localises at the outer periphery of the nuclear membrane. *Fkb-6* mutant worms exhibit decreased chromosome pairing and non-homologous synapsis as well as meiotic DNA damage repair defects. FKB-6 has been proposed to regulate microtubule dynamics in the germline and to downregulate dynein activity. This would prevent excess chromosome movement, alleviating the nucleus transiently, and thus allowing time for the homologues to recognize each other [69]. MRG-1 is the homolog of the mammalian chromodomain protein MRG15 [70]. In *C. elegans*, it is associated to autosomes and is essential for the proliferation of primordial germ cells as well as for X-chromosome silencing in the germline [84]. During gametogenesis, *mrg-1* mutants display defects in pre-synaptic pairing, synapsis and double-strand break repair [71,72]. While the depletion of MRG-1 function does not affect the pairing of PCs, this protein is essential to the proper alignment of non-PC loci and important to ensure that non-homologous synapsis does not occur [71]. PPH-4.1 is a widely conserved serine/threonine phosphatase whose depletion induces a reduction of autosomal pairing, synapsis between non-homologous chromosomes, and defects in double-strand break formation and repair. The protein PPH-4.1 has been proposed to regulate the phosphorylation of several proteins during gametogenesis, in particular PLK-2, SUN-1 and/or of components of the synaptonemal complex [73].

The polymerisation of the central region of the synaptonemal complex between homologous axes initiates close to (but not exactly at) the PC, and likely at loci located more distal to the PCs [85]. The SC polymerises progressively in the transition zone, starting as stretches associated with coalescing axes of aligned homologous chromosomes. Synapsis of the X chromosomes takes more time than the autosomes, but the reasons for this remain unknown [30]. Despite the fact that they are loci of synapsis initiation, PC regions are not essential to SC polymerisation. In mutant contexts, synapsis can even occur between homologous sequences at PC regions, while being non-homologous at distal loci [71]. Polymerisation of the central region of the SC between chromosome axes thus appears to rely more on the proximity between two axes and the physical properties of its components rather than on sequence homology [86]. PC pairing and SUN/KASH/cytoskeleton-driven movements ensure that non-homologous associations are disfavored and destabilized.

Upon entry in pachytene, when full synapsis is achieved on every chromosome, chromatin clustering and SUN/KASH-complex aggregation are relieved, PLK-2 relocates to the SC, and autosome movements are reduced. Only the X chromosomes remain mobile for a longer period [42]. Whilst the concomitance of a fully polymerised SC and chromosome movement arrest suggest a link between these two events, the signal transducer remains unknown. The dynamic localization of PLK-2 during early meiosis could be involved [68], possibly through a feedback signal of SC components on CHK-2 upon the completion of pairing and synapsis [87].

### 1.2. Drosophila Melanogaster

Surprisingly, it is only recently that meiotic chromosome pairing has been studied at the cytological level in *Drosophila* females, despite more than a century of genetic studies. One reason was the observation that the four pairs of homologous chromosomes are paired in all somatic cells studied in *Drosophila*. A phenomenon called *somatic* pairing is common to many dipteran. It was thus expected that meiotic pairing in germ cells was just a continuation of somatic pairing [88,89,90,91,92]. However, we and others have shown that chromosomes are not paired in embryonic PGCs and that this unpaired state is maintained into adult germline stem cells (GSCs) [15,16]. Of all the cells in the adult flies, the only cells with unpaired chromosomes are the early germ cells conceived to go through meiosis. In *Drosophila* females, mitosis and meiosis occur sequentially throughout adult life in two distinct regions of the germarium, at the tip of each ovary [93]. In the mitotic zone (called region 1), GSCs generate a precursor cell called a cystoblast (CB), which undergoes exactly four mitoses to produce a germline cyst made of 16 cells. These mitoses are not complete, and all 16 sister cells remain connected through ring canals and by an organelle called the fusome, which links all cells. After the last mitosis, 16-cell cysts enter region 2, and all 16 cells start meiosis [94]. During differentiation in region 2, only one cell per cyst, however, remains in meiosis, while the 15 others exit meiosis and endoreplicate their DNA [95]. We and others found that homologous chromosomes first pair through centromeres and euchromatic loci. This early pairing occurs progressively during the four mitotic cycles before the entry in meiosis and the formation of DSBs [15,16]. It is thus likely that centromere pairing is independent of DSBs. This is in agreement with previous studies showing that homologues are already paired when entering meiosis [15,16,31,32]. Interestingly, this premeiotic pairing requires components of the synaptonemal complex, such as C(3)G and Corona, which localise at centromeres [15]. It is also similar to the initiation of meiosis in budding yeast, where centromeres also become “coupled” early in meiotic prophase [4]. This early association depends on Zip1, a central component of the SC functionally similar to C(3)G in flies and SYCP1 in mice. Similarities further extend to the localisation of Zip1, which partially overlaps with yeast centromeres at this early stage, like C(3)G and CenpA in flies [4]. In contrast, the X chromosome behaves differently as it appears always paired at its centromeric regions throughout oogenesis [15]. This pairing is independent of C(3)G and Corona, and the underlying mechanisms remain to be identified. Since both X-chromosomes have centromeric clusters of rDNA repeats that localise to the same nucleolus, we hypothesised that mechanisms similar to *Drosophila* males X-Y pairing might contribute to pairing in this centromeric region [96]. It is thus likely that additional mechanisms also facilitate the pairing of autosomal homologues in germline cysts.

More recently, we found that one such mechanism, which facilitates chromosome pairing, is nuclear rotation driven by microtubules [39]. By tracking the movements of centromeres and the nuclear envelope, we observed that nuclei became highly dynamic, undergoing cycles of rotations. We measured that centromeres move at an average speed of 300 nm/sec. These rotations are driven by microtubules nucleated mainly from the fusome, but also from the nuclear envelope and from centrosomes. The microtubule motor Dynein is required, as no rotation is observed in *dynein* mutant germ cells. Consequently, homologues fail to pair and SC formation is greatly compromised. Like *C. elegans*, cytoskeletal forces are transmitted to the nuclear envelope and chromosomes by the LINC complex, made of the SUN-domain protein Klaroid and the KASH-domain protein Klarsicht. Both Klaroid and Klarsicht localise as dots at the nuclear envelope and colocalise with centromeres. In the absence of the LINC complex, rotations are severely slowed down, but some movements still occur. In these conditions, homologous chromosomes managed to pair to some extent, in a very delayed timing, which disrupted the formation of the SC. Similarly, in mutant germ cells without centrosomes, rotations were reduced and the initial pairing of homologues was disrupted. However, the clustering of centromeres in older cysts was only mildly affected. It will be interesting to investigate what drives the movements of chromosomes in addition to nuclear rotations. 

Next, the four pairs of centromeres cluster to form one or two masses when entering meiosis [31,32] (reminiscent of the chromocenter of polytene chromosomes in salivary glands [97]). The clustering of paired centromeres requires SC components but does not depend on DSB formation. Polymerisation of the SC starts at centromeres, followed by several euchromatic sites and then at many more loci along chromosomes. Telomeres do not form a bouquet in *Drosophila* [31,94]. Centromeres clustering may thus be the functional equivalent of the bouquet in other species. 

Conformation capture technology such as Hi-C has revealed that the fly genome is organised into topologically associated domains (TADs). These chromatin territories may represent functional units of the genome on each chromosome. Recently, TADs were also proposed to be high-affinity pairing sites *between* homologous chromosomes in somatic cells [98]. TADs would act like “buttons” driving close pairing of homologues. Transgenes containing entire TADs were shown to be sufficient to initiate homologous pairing. It will be very interesting to test if such a model applies to meiotic pairing in germ cells. 

### 1.3. Danio Rerio

Excellent genetics and genomic tools have made the success of the zebrafish (*Danio rerio*) as a complementary model organism to *C. elegans*, *Drosophila* and mammal model organisms. Regarding meiosis, females produce oocytes throughout their adult life, ensuring a steady production of the different meiotic stages. The zebrafish karyotype is made of 25 pairs of chromosomes. Most chromosomes are subtelomeric and submetacentric, and only two are metacentric [99]. It is only recently that the timing of meiotic progression has been described systematically in males and females [17]. It was found that the formation of a telomere bouquet was a key event of early meiosis in zebrafish. Indeed, DSB formation, homologues pairing and synapsis, all start at, or close to, telomeric ends, when telomeres cluster in leptotene/early zygotene. Thus, there is a dramatic polarisation of DSBs and synapsis at telomeres [17,100]. Synapsis then proceeds from telomeres inward to the rest of the chromosome. Analysis of *spo11* mutant zebrafish showed that the initial formation of the bouquet does not depend on Spo11. However, the subsequent co-alignment of homologous chromosomes and the formation of the SC does depend on DSBs. Furthermore, DSBs appear and localise near telomere ends just before the onset of pairing and synapsis at these same loci. This timing and localisation support the idea that DSBs initiate pairing and synapsis in zebrafish. A second wave of DSBs along chromosomes may help pairing and synapsis of interstitial sites between homologues at later stages.

The pattern and timing of DSBs, pairing and synapsis initiation are very similar in male and female zebrafish. However, there are some interesting differences. In males, the localisation of crossing-overs is marked by Mlh1 and follows DSB polarisation toward the bouquet [101]. In contrast, in females, Mlh1 is more evenly distributed along chromosomes. This difference translates into dissimilarities in the genetic map with recombination biased toward telomeres in males but not in females [102,103]. In addition, *spo11* mutant males are completely sterile, whereas mutant females produce some viable oocytes. The progeny of these mutant females is, however, often abnormal. The causes of these differences remain to be explored.

Studies on fixed samples set the stage for further analysis of chromosome movements and dynamics [17,100]. Microtubule inhibitors are known to disrupt both the formation of the bouquet and of the Balbiani body (a structure containing mitochondria, ER and Golgi vesicles) [47]. Microtubules may thus play an important role in pairing and synapsis of chromosomes in zebrafish. Taking advantage of transparency of gonads, future studies will surely reveal the dynamics of chromosomal movements and pairing of homologues in zebrafish meiosis.

### 1.4. Mus Musculus

In mouse, primordial germ cells (PGCs) are derived from a small number of epiblast cells. Once specified, PGCs migrate across the embryo to reach the developing genital ridge (GR) around E10.5 [104], where they proliferate rapidly [105,106]. In female embryos, PGCs proliferate until ∼E13.5. Then, they enter meiotic prophase I [107,108] until they arrest at the diplotene stage of meiotic prophase I. Hormonal stimulation triggers the completion of the first meiotic division, while the second meiotic division happens after fertilization. Since early meiotic stages from leptotene to pachytene are embryonic, it is technically difficult to observe these stages in oocytes by live imaging, unless they have been prepared as two-dimensional spreads. In male embryos, PGCs stay quiescent for the remaining embryonic development [107,109]. At day 5 postpartum (P5), many PGCs resume active proliferation and some become spermatogonial stem cells (SSCs) [110,111]. Since prophase spermatocytes can be massively collected from juvenile male mice without microdissection, chromosome dynamics and synapsis have thus been extensively studied in males rather than in females.

Recent transcriptomic analyses have shown that meiotic genes involved in prophase I are expressed and translated long before the initiation of the meiotic process. For example, REC8 and synaptonemal complex proteins are expressed in spermatogonia, which go through several mitotic divisions before meiotic entry [112,113]. This is similar to the expression of C(3)G in *Drosophila* during pre-meiotic stages. Interestingly, homologue associations have also been detected very early in mouse spermatocytes, before the appearance of DSBs. One study observed evidence of pairing as early as the pre-meiotic S-phase [22], while two other studies detected pairing in early leptotene [23,24]. They both agreed that these early pairing events are DSB-independent. However, Boateng and colleagues found that it was dependent on *Spo11*, but independent of SPO11 catalytic activity. In contrast, Ishiguro and colleagues found a requirement for RAD21L, a meiotic-specific cohesin, but not for SPO11. This requirement seems stronger in males than in females, as *Rad21L* mutant oocytes progressed to pachytene with a substantial number of chromosomes paired. Despite some differences, these studies revealed that homologues associate much earlier than previously thought in male mice, which is consistent with data obtained in *S. cerevisiae* and *Drosophila*. However, this has not been observed in females [19]. This “early” pairing of homologues is later stabilised by recombination events and the formation of the SC [114]. In females, synapsis initiation is biased toward distal and interstitial regions of chromosomes, whereas centromeric regions appear to synapse much later [19]. Furthermore, SC initiation occurs in oocytes with little or no sign of bouquet [19]. These conclusions are based, however, on fixed oocytes, and the presence of a bouquet may be highly dynamic and only visible by live-imaging at this stage (see [115]). Nevertheless, in both males and females, these early signs of homologous chromosome associations question the role of the bouquet for homologous synapsis. The bouquet is only transiently observed at leptotene/zygotene in males and for a bit longer during zygotene/pachytene in oocytes. In addition, several studies showed that pairing occurs at many interstitial sites along the entire chromosome length (except at centromeres) and not only at telomeres [22,23,116,117]. A possible function for the bouquet would be to promote pairing in regions not paired during the previous leptotene stage, such as the peri-centromeric domains. 

Direct observations of chromosome movements and pairing are challenging in male spermatocytes, but it is even more difficult in oocytes because early meiosis occurs during fetal development. Most published studies thus analysed spermatogenesis, and here, we will compare these results with chromosome dynamics in oocytes when data are available in females. In mouse spermatocytes, chromosomes display oscillations from late leptotene to early pachytene with the fastest movements happening at the zygotene stage (with speed ranging from 24 to 130 nm/sec) [35]. These fast movements are a combination of rotations of the entire nuclear envelope and individual chromosome movements. The bouquet stage is thus not the stage with the highest velocity. To observe oocyte meiotic chromosomes, transgenic mice expressing N- or C-terminal fluorescent-tagged SYCP3 (lateral element of SC) were generated to label the SC [115]. As in males, two types of movements were observed in prophase oocytes: nuclei rotations and individual chromosomes movements at the nuclear envelope. At zygotene, oocytes were observed rotating for long periods of time in a single direction, whereas spermatocytes displayed wiggling movements at pachytene. In males, clustered telomeres move less compared to stages preceding or following bouquet formation [34]. Centromeres also remain close to the nuclear periphery at early leptotene and cluster with the bouquet [18]. These telomere movements rely on the microtubule cytoskeleton and microtubule-associated Dynein [34,35]. A complex microtubule network associated with the outside nuclear envelope has been described in spermatocytes by cytology [34,35]. These microtubule cables are organised dynamically around the nucleus, depending on the stage of prophase I [35]. Observations of trajectories have revealed that several chromosomes can move along the same path; one chromosome can also move back and forth along a particular trajectory displaying frequent directional changes [35]. Whether a similar microtubule network exists in oocytes remains to be determined. Live-imaging in oocytes, nonetheless, has revealed that telomeres cluster as a bouquet during mid- to late zygotene, confirming previous observations made on fixed samples. It further showed that this conformation is highly dynamic with rapid cycles of bouquet formation and dissolution. In some instances, these cycles appear synchronised between neighboring oocytes, suggesting that they could be sister oocytes linked by cytoplasmic bridges and part of a single syncytium [118].

As in other species, microtubules forces are transduced to chromosomes by the LINC complex localised at the nuclear envelope. In mouse, the complex is composed of SUN1, forming attachment sites for telomeres at the inner nuclear membrane (INM), and the meiotic specific KASH5, linked to Dynein, at the outer nuclear membrane (ONM). Both SUN1 and KASH5 are also found in oocytes. SUN2, another SUN domain protein also found in oocytes, partially compensates for SUN1 function in both males and females in *Sun1* null mutants [62]. SUN1 and SUN2 are distributed evenly around the nuclear envelope at leptotene [61,62]. At later stages, SUN1 forms aggregates, which colocalise with telomeres at the nuclear envelope [61]. Whether SUN2 displays the same pattern as SUN1 in oocytes is unknown [119]. KASH5 also forms dots and localises next to telomeres [63]. In *Sun1* mutant oocytes, telomere attachment to the nuclear envelope is lost, and SC formation is greatly impaired. Telomere movements disappear in *Kash5* mutant males and are strongly impaired in *Sun1* mutant males [35]. Both mutant conditions lead to pachytene arrest and massive apoptosis of spermatocytes [61,120].

In both spermatocytes and oocytes, great advances have been made to understand how telomeres are recruited to the LINC complex. Despite several differences between males and females, most molecular components are shared. At early prophase I, TERB1 links SUN1 and the telomere binding protein TRF1 [53,54]. At mid-prophase I, the TERB2–MAJIN complex binds telomeres through the MAJIN DNA binding domain [55]. TERB2 and MAJIN are required for TERB1 recruitment to LINC [55]. Interestingly, MAJIN has a trans-membrane domain (in addition to its DNA binding domain) that may help to stabilise the LINC-TERB1 complex at the nuclear membrane [55]. In males, CDK2 (cyclin-dependent kinase 2) has been proposed to regulate the LINC complex by phosphorylating TERB1 [55]. On the other hand, CDK2 may regulate the interaction between the LINC complex and SUN1 [74]. Speedy/RINGO, a non-canonical activator of CDKs, was shown to be important for CDK2 localisation at telomeres in mouse oocytes and spermatocytes, suggesting that a telomeric pre-complex includes Speedy/RINGO and CDK2. [74,75]. Although telomeric proteins are rarely conserved between species, this molecular framework deciphered in mouse will certainly help investigate similar processes in other species. 

## 2. Concluding Remarks

The main purpose of the oocyte is to transmit meiotic chromosomes to the next generation. During the making of the oocyte, chromosomes arguably go through the most complex and dramatic events of any cell type. Here, we described an exciting diversity of mechanisms to reach that goal, but common themes are also emerging among metazoan. For instance, microtubules and the LINC complex seem to facilitate chromosomes dynamics and pairing in all species studied so far. These movements are also needed to disentangle and to remove inappropriate interactions in all species. The geometry of chromosome movements varies, however, in different species. It can follow global rotations or oscillations of entire nuclei or random movements of single telomeres. These principles also extend to plants. In maize zygotene nuclei, telomeres are attached to the inner nuclear membrane and form a bouquet. Live-imaging studies found that the large chromosomes of maize undergo coordinated, rapid and abrupt telomere-led movements [121]. In *Arabidopsis*, telomeres also cluster as a bouquet, and SUN-homolog mutants have a delayed meiotic progression, defective SC formation, and decrease in chiasmata [122]. These results suggest that chromosome movements are probably important for meiosis in *Arabidopsis* too, although not yet directly observed by live-imaging.

How these movements are regulated remains mostly unknown. What triggers chromosome motion? Additionally, what stops it when chromosomes have fully synapsed? It may involve some sort of bi-directional communications between inside the nucleus and the cytoplasm. This feedback remains to be identified. As we described here, chromosome movements depend on microtubule organization, which itself depends on the organisation and polarisation of the cell. How cell polarity relates to nuclear polarity will be a very interesting field of research. For example, in mouse and *Drosophila*, microtubule-driven nuclear rotations are required for homologues pairing. What organization of microtubules and motor proteins underlies these rotations in a constant direction? In the zebrafish, the Balbiani body is a microtubule-organizing center, which polarises the oocyte. How does the polarisation of the Balbiani body relate to the formation of the bouquet?

There are also interesting differences between males and females within the same species. In *Drosophila*, males do not induce DSBs or form SC, but still manage to pair and segregate their chromosomes. In many species, the X chromosome also behaves differently to the autosomes. In *Drosophila* females, X chromosomes appear always paired, while they are the last ones to pair in *C. elegans.* The reasons and molecular mechanisms underlying this sexual dimorphism are mostly unknown and are exciting future studies. The advance of genome engineering technologies, such as CRISPR/Cas9, will allow us to investigate meiosis mechanisms in a wider range of species from different branches of the tree of life. The more species studied, the more general principles will emerge of how to manufacture an oocyte.

## Figures and Tables

**Figure 1 cells-09-00696-f001:**
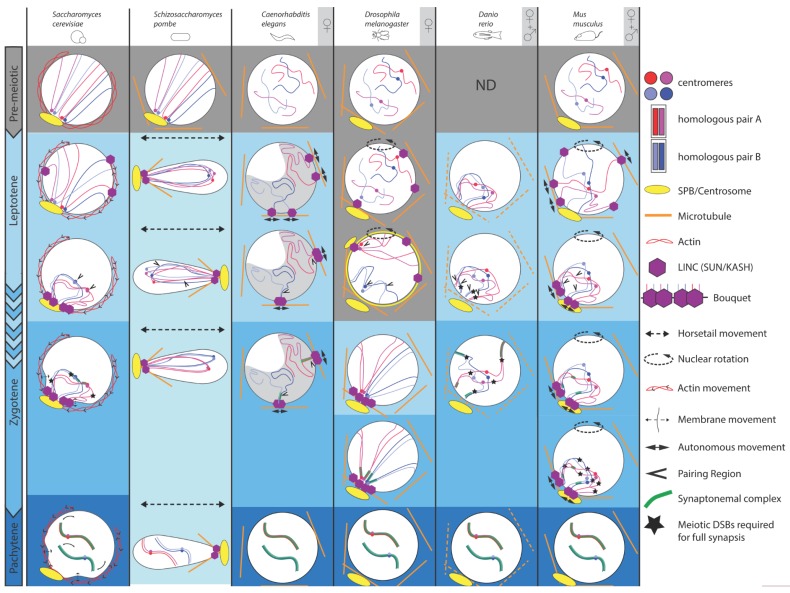
Comparison of chromosomal and nuclear meiotic movements in different species. Yeasts, *C. elegans*, *Drosophila*, zebrafish and mice exhibit dramatic changes throughout meiotic prophase I substages, i.e., leptotene, zygotene, pachytene and diakinesis (not shown). These substages are well defined, except for *S. pombe*, which has no synaptonemal complex (SC; lightest blue), but displays peculiar LINE structures reminiscent of SC. For our selected sexually reproducing models, data were obtained either only in females ♀ (oogenesis), or also from males (♀ + ♂) (spermatogenesis). At pre-meiotic stages (grey background) in yeasts and mouse, Rabl orientation is inherited from previous divisions, when centromeres (blue and red dots) are pulled toward the spindle pole body or centrosome (yellow oval with emanating microtubules), with trailing telomeres. *C. elegans*, *Drosophila* and mouse do not show such organization. Zebrafish chromosomes organization has not been defined at this stage. Chromosome dynamics start from early leptotene, except in *Drosophila*, where movements take place during premeiotic divisions (grey background). Nuclear movements (dashed arrows) occur in yeasts, *Drosophila* and mouse, thus generating coordinated chromosome dynamics, whereas solitary chromosome movements (plain arrows) occur in *C. elegans* and, in addition to coordinated movements, in mouse. These movements rely on the coordinated action of the microtubule cytoskeleton (except in *S. cerevisiae,* actin), the dynein motor and the LINC complex (SUN/KASH). In *S. cerevisiae*, early movements depend on actin, whose rapid polymerization cycles push on the nucleus, consequently shaking chromosomes. At pachytene, telomeres display abrupt movements concomitant with nuclear membrane deformations. In *S. pombe*, prophase I-like stage is called the horsetail stage: the nucleus elongates and moves back and forth between the ends of the cell. Telomeres remain clustered at the leading edge of the moving nucleus. *Drosophila* and mouse display entire nuclear rotations driven by the microtubule cytoskeleton. Solitary microtubules-driven movements (plain arrows) have also been identified in *C. elegans* through the coordinated action of microtubules, dynein and LINC complex at the nuclear membrane. Concomitantly with early movements, centromeres move away from the pole, while telomeres attach to the nuclear membrane and move to a small area adjacent to the spindle pole, forming a bouquet at leptotene/zygotene transition in yeast and mouse, or early leptotene in zebrafish. *C. elegans* oocytes pack their chromatin in a characteristic half-moon territory. Resulting proximity of specific chromosomal regions (see text) leads to initial homologous pairing (arrowheads). Except for zebrafish, initial pairing is DSB-independent. Entry in zygotene is marked by the initiation of the synaptonemal complex formation (green ladder-like structure). During zygotene, chromosomes move out of the bouquet. Completion of synapsis and resolution of interlocks marks pachytene, displaying well-separated chromosome pairs. Full synapsis of homologues requires DSBs, except in *C.elegans* and *Drosophila*.

**Table 1 cells-09-00696-t001:** Comparison of chromosomes and nuclear meiotic movements in different species.

	*S. Cerevisiae*	*S. Pombe*	*C. Elegans*	*D. Melanogaster*	*D. Rerio*	*M. Musculus*
**Chomosome number (2n)**	32	6	12	8	50	40
**Localisation of initial homologous pairing**	Centromere [4,5]	Arms [6,7] Centromeres [8,9,10]	Pairing Center [11,12,13,14]	X-Y rDNA Centromeres [15] Euchromatic Loci [15,16]	Sub Telomere [17]	♂ Telomeres [18] ♀ Distal/Interstitial Regions [19]
**Timing of initial homologous pairing**	Zygotene [4]	Prophase [20]	Transition Zone [21]	Mitotic Region [15,16]	Leptotene/Early Zygotene [17]	Premeiotic S-phase [22]/Early Leptotene [23,24]
**Meiotic DSB dependent synapsis**	Yes [25]	NA	No [26]	No [27]	Yes [17]	Yes [22,23,24]
**Type of bouquet**	Telomeres [28]	Telomere [20]	Half-moon Shape [21,29,30]	Centromere Cluster [31,32]	Centromere Telomere [17]	Centromere [18,33] Telomere [34]
**Bouquet stage**	Lept/Zyg Transition [28]	Prophase [20]	NA	NA	Leptotene [17]	♂ Lept/Zyg Transition [34,35] ♀ Zyg/Pachytene Transition [19]
**Chromosome** **movement type**	Coordinated-[36,37] autonomous [37,38]	Horsetail Movements [20]	Autonomous [30]	Nuclear Rotation [39]	ND	Nuclear Rotation Autonomous [35]
**Chromosome** **movement stages**	Prophase [36,40]	Horsetail Stage [20]	Leptotene-zygotene [41,42]	8-cell cyst [39]	ND	Leptotene-zygotene [35]
**Force-inducing cytoskeleton**	Actin [36,43,44]	Microtubules [45]	Microtubules [46]	Microtubules [39]	Microtubules? [47]	Microtubules [34,35]
**Speed (nm/sec)**	300–500 [37,44]	83 [20]	125–400 [41,42]	300 [39]	ND	109–120 [35]
**Adaptors to nuclear envelope**	Ndj1p/Csm4p [36,43,44]	Taz1/Rap1 Bqt1/2 [48,49,50]	HIM-8/ZIM-2/ZIM-1/ZIM-3 [51,52]	ND	ND	TERB1-TRF1 [53,54] TERB2-MAJIN [55].
**LINC**	Mps3p/Mps2p [44,56,57]	Sad1/Kms1/Kms2 [58,59,60]	SUN-1/ZYG-12 [46]	Klarsicht, Klaroid [39]	ND	SUN1/SUN2/ [61,62] KASH5 [63]
**Motors**	ND	Dynein [48,49,50]	Dynein [46]	Dynein [39]	ND	Dynein [34,35]
**Other regulators**	ND	MAP/Pat1 [20,50,64]	CHK-2/PLK-2 [65,66]/HAL-2/HAL-3 [67,68] FKB-6 [69] MRG-1 [70,71,72] PPH-4.1 [73]	ND	ND	Rad21 [23] CDK/Cdk2 [74]/ SpeedyA [74,75].

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
