# Peer review of "Mixing and Matching Chromosomes during Female Meiosis"

_cells, 2020, doi:10.3390/cells9030696_

Round 1

Reviewer 1 Report

This is an ineresting review on a subject that is not well covered.

It is globally well written and I have only a few suggestions.

Page 1, first paragraph. It is unclear which organisms are covered (animals I believe). Does the germline cysts with incomplete mitosis exists in all animals?

page2. "spo11 is conserved in alamost all species". Almost? references?

Line 81. It should be made clear from the begining that there is no SC in pombe.

Line 108. To help the naive reader, please define "hollocentric"

L242. "Invertebrate" in an obsolete term, because this is not a monophyletic clade. What do you actullay mean? Animals? Droso and C.elegans?

Author Response

Page 1, first paragraph. It is unclear which organisms are covered (animals I believe). Does the germline cysts with incomplete mitosis exists in all animals?

Our manuscript covers mostly animals, but we mention yeasts in the introduction and plants in the conclusion. Germline cysts exist in most animals but not all. In particular, there is no germline cysts in female c. elegans. we thus decided to remove this sentence. it was replaced with the sentence: "Depending on the species, these precursors can either become stable germline stem cells or differentiate by undergoing a limited number of mitosis. "

page2. "spo11 is conserved in alamost all species". Almost? references?

This has been changed to "Meiotic DSBs are induced by the topoisomerase-like Spo11, which is conserved in all species studied so far. "

Line 81. It should be made clear from the begining that there is no SC in pombe.

Done. Now line reads "The fission yeast, Schizosaccharomyces pombe, has three pairs of chromosomes and does not form a synaptonemal complex"

Line 108. To help the naive reader, please define "hollocentric"

Line 11 now reads: "C. elegans has six pairs of holocentric chromosomes, i.e. which centromeres are distributed along the entire chromosome length. "

Sentences were also reworked lines 106 to 113 to make more sense to the reader.

L242. "Invertebrate" in an obsolete term, because this is not a monophyletic clade. What do you actullay mean? Animals? Droso and C.elegans?

Done. Line 252 now reads: "Excellent genetics and genomic tools have made the success of the zebrafish (Danio rerio) as a complementary model organism to C. elegans, Drosophila and mammal model organisms"

Reviewer 2 Report

This paper reviews the dynamics of chromosomes leading to homologous pairing, synapsis and recombination in female meiosis of four different organisms: Caenorhabditis elegans, Drosophila melanogaster, zebra fish and mouse. Features of early stages of female meiosis in such organims are described in detail. However, after caerful reading of the work, attracts attention that no mention is made of female meiosis in plants such as Arabidopsis and that possible connections with human female meiosis are not considered. In my opinion, an appropriate development of these two points would considerably increase the scope of the review, which will become useful for a greater number of reader.

eatures 

Author Response

This paper reviews the dynamics of chromosomes leading to homologous pairing, synapsis and recombination in female meiosis of four different organisms: Caenorhabditis elegans, Drosophila melanogaster, zebra fish and mouse. Features of early stages of female meiosis in such organims are described in detail. However, after caerful reading of the work, attracts attention that no mention is made of female meiosis in plants such as Arabidopsis and that possible connections with human female meiosis are not considered. In my opinion, an appropriate development of these two points would considerably increase the scope of the review, which will become useful for a greater number of reader.

We agree with the reviewer that works performed in plants have considerably advanced our understanding of meiosis in general. However, live-imaging of chromosomes remains very technically difficult in plants and only a few studies have been published. We also felt that it was important to mention these studies. We thus have added in the concluding remarks Line 387:"These principles extend also to plants. In maize zygotene nuclei, telomeres are attached to the inner nuclear membrane and form a bouquet. Live-imaging studies found that the large chromosomes of maize undergo coordinated, rapid and abrupt telomere-led movements [122]. In Arabidopsis, telomeres also cluster as a bouquet, and SUN-homolog mutants have a delayed meiotic progression, defective SC formation, and decrease in chiasmata [123]. These results suggest that chromosome movements are probably important for meiosis in Arabidopsis too, although not yet directly observed by live-imaging."

Reviewer 3 Report

This review article covers extensively the topic of homolog pairing during the prophase of meiosis, promoted by chromosome movements, with a special emphasis on the oocyte, and comparing several model organisms, starting from the two yeasts, S. cerevisiae and S. pombe, and then reviewing in detail the worm C. elegans, the fly Drosophila, zebrafish and the mouse models.

A table and a figure comparing the different systems is provided.

This review is timely and provides many useful pieces of information. However, I found it lacks a bit of focus and is rather descriptive. Maybe the final paragraph could be expanded to better summarize the similarities and differences between the experimental systems since they are described separately along the review.

I also think that a general scheme of meiotic chromosome segregation would be useful as an introduction in addition to the existing figure 1.

For the Table 1, to facilitate reading, the empty slots where information is missing should be filled with either “NA” (not applicable) or “ND (not determined). In addition, since the table is quite long, the column titles should be present at each beginning of a new page.

In many instances in the text, italics are missing, for instance for C. elegans. This should be corrected. I also found several typos that should be re-read carefully.

More specific comments:

  • Introduction: the term “synapsis” should be described earlier to avoid confusion with “pairing”. It could be first mentioned on the first page, upon the first description of the SC: “Pairing is then stabilized by the polymerization …called the synaptonemal complex (SC) all along homologous chromosomes (synapsis).”
  • Introduction, beginning of page 2: Spo11 is a “topoisomerase-like” protein
  • Table 1:
    • For “Meiotic DSB dependent synapsis”: for S. pombe: it is not correct to talk about synapsis, because this organism does not form a synaptonemal complex. Better to indicate here “NA”, even if the pairing is DSB-independent (which is already mentioned in the text).
    • For the category named “bouquet”: do the authors here mean “regions involved” or “type of bouquet”? This would be more meaningful than just “bouquet”.
    • “timing of initial homologous synapsis”: for D. renio, write leptotene/early zygotene instead of “L/EZ”
    • For the “LINC” category: the reference for the last paper of R. Pezza identifying Mps2 as the link between telomeres and the actin cytoskeleton should be added (Lee et al 2020 Current Biology)
  • Page 6 lines 24 and 26: these two sentences are contradictory: they seem to indicate that mouse chromosomes are first organized in a Rabl configuration, and then say that mouse chromosomes do not undergo this configuration. This should be corrected.
  • Page 7 2nd line: it is not clear if mouse chromosomes are experiencing both solitary movements in addition to coordinated movements. If this is the case, this should be phrased more clearly. (something like “in addition to /following coordinated movements”).
  • Page 7 line 46: this sentence is not clear. Do the authors mean “Initial / early pairing is DSB-independent”?
  • Page 7 line 62: as indicated for the Table 1, the recent paper by R. Pezza lab should be cited here (Lee et al 2020 Current Biology) for the role of Mps2.
  • Page 9 line 144: please specific that you are talking about the C. elegans meiotic polo kinases (PLK-2 and PLK-1).
  • Page 9 line 173: “only the X chromosomes remain mobile”
  • Page 10 line 196: the authors should have mentioned in the previous sentence that the early pairing involves the centromeres (line 194 when describing the timing of pairing).
  • Page 10 line 200: correct to: “also become coupled early in meiotic prophase”
  • Page 10 line 210: this sentence is not grammatically correct. “one such mechanism…are nuclear”
  • Page 10 line 215: “no rotation is observed”
  • Page 10, line 227: “in a second step,”: it is not clear which first step this is following. It could be clarified.
  • Page 12, line 287: meiosis is not a cell cycle. Maybe replace by “the meiotic process”.
  • Page 289: to my knowledge, in S. cerevisiae, Zip1 is not expressed in premeiotic cells. It is induced during the meiotic prophase, at the same time as the meiotic recombination genes, even though it localizes of centromeres prior to / in dependently of recombination initiation. This should be corrected.

Author Response

 Maybe the final paragraph could be expanded to better summarize the similarities and differences between the experimental systems since they are described separately along the review.

Following the reviewer comment, we have re-organized our concluding paragraph. In particular, we extended our conclusions to plants, and quickly described studies using plants as a model system, Line 387.

I also think that a general scheme of meiotic chromosome segregation would be useful as an introduction in addition to the existing figure 1.

We have carefully thought about this suggestion, however we felt that there are already numerous reviews illustrating these early steps of meiosis with very clear figures.

For the Table 1, to facilitate reading, the empty slots where information is missing should be filled with either “NA” (not applicable) or “ND (not determined). In addition, since the table is quite long, the column titles should be present at each beginning of a new page.

This has been corrected from our end. we hope that format will be conserved by during the journal editing process.

Introduction: the term “synapsis” should be described earlier to avoid confusion with “pairing”. It could be first mentioned on the first page, upon the first description of the SC: “Pairing is then stabilized by the polymerization …called the synaptonemal complex (SC) all along homologous chromosomes (synapsis).

This has been changed to "Pairing is then stabilized by the polymerization of a proteinaceous scaffold called the synaptonemal complex (SC), which holds together homologous axes (synapsis), and promotes genetic recombination. "

Introduction, beginning of page 2: Spo11 is a “topoisomerase-like” protein

Done

For “Meiotic DSB dependent synapsis”: for S. pombe: it is not correct to talk about synapsis, because this organism does not form a synaptonemal complex. Better to indicate here “NA”, even if the pairing is DSB-independent (which is already mentioned in the text).

This has been corrected

For the category named “bouquet”: do the authors here mean “regions involved” or “type of bouquet”? This would be more meaningful than just “bouquet”.

We changed it to "Type of Bouquet"

“timing of initial homologous synapsis”: for D. renio, write leptotene/early zygotene instead of “L/EZ”

Done

For the “LINC” category: the reference for the last paper of R. Pezza identifying Mps2 as the link between telomeres and the actin cytoskeleton should be added (Lee et al 2020 Current Biology)

We thank the reviewer for pointing this reference and this reference has been added. 

Page 6 lines 24 and 26: these two sentences are contradictory: they seem to indicate that mouse chromosomes are first organized in a Rabl configuration, and then say that mouse chromosomes do not undergo this configuration. This should be corrected.

It has been changed to: "At pre-meiotic stages (grey background) in yeasts and mouse, Rabl orientation is inherited from previous divisions, when centromeres (blue and red dots) are pulled toward the spindle pole body or centrosome (yellow oval with emanating microtubules), with trailing telomeres."

Page 7 2nd line: it is not clear if mouse chromosomes are experiencing both solitary movements in addition to coordinated movements. If this is the case, this should be phrased more clearly. (something like “in addition to /following coordinated movements”).

It has been changed to:"Nuclear movements (dashed arrows) occur in yeasts, Drosophila and mouse, thus generating coordinated chromosome dynamics, whereas solitary chromosome movements (plain arrows) occur in C. elegans and, in addition to coordinated movements, in mouse. "

Page 7 line 46: this sentence is not clear. Do the authors mean “Initial / early pairing is DSB-independent”?

This has been changed to "initial".

Page 7 line 62: as indicated for the Table 1, the recent paper by R. Pezza lab should be cited here (Lee et al 2020 Current Biology) for the role of Mps2.

We thank the reviewer for pointing this reference and this reference has been added. 

Page 9 line 144: please specific that you are talking about the C. elegans meiotic polo kinases (PLK-2 and PLK-1).

Indeed there are several PLKs in C. elegans. This was specified.

Page 9 line 173: “only the X chromosomes remain mobile”

now line 183: "Only the X chromosomes remain mobile for a longer period [44]"

Page 10 line 196: the authors should have mentioned in the previous sentence that the early pairing involves the centromeres (line 194 when describing the timing of pairing).

Now line 205: "We and others found that homologous chromosomes first pair through centromeres and euchromatic loci. "

Page 10 line 200: correct to: “also become coupled early in meiotic prophase”

Done

Page 10 line 210: this sentence is not grammatically correct. “one such mechanism…are nuclear”

Done Line 222: "More recently, we found that one such mechanism, which facilitates chromosome pairing is nuclear rotation driven by microtubules [41]. "

Page 10 line 215: “no rotation is observed”

Corrected

Page 10, line 227: “in a second step,”: it is not clear which first step this is following. It could be clarified.

It has been changed to "Next,..." line 238

Page 12, line 287: meiosis is not a cell cycle. Maybe replace by “the meiotic process”.

It has been changed to "the meiotic process"

Page 289: to my knowledge, in S. cerevisiae, Zip1 is not expressed in premeiotic cells. It is induced during the meiotic prophase, at the same time as the meiotic recombination genes, even though it localizes of centromeres prior to / in dependently of recombination initiation. This should be corrected.

To avoid any confusion, the reference to zip1 in that context has been removed and only the reference to C3G remains.

Reviewer 4 Report

Authors of review, Mixing and matching chromosomes during female meiosis, describe the chromosome movements, positioning and the key players involved at the pre-prophase I of female meiosis. Authors also describe principal similarities and differences of this process for the most used animal models used for the studies. The review focuses specially on the mechanisms of chromosomes movement to be properly organised with their respective homologs. The authors included information from well accepted findings beside of novel findings in the field.

Despite that authors worked with literature extensively, some parts of the text have not been referenced and lack citations (see comments below).

The main idea of the mechanisms underlying chromosome movement at this stage of meiosis (as well as the importance of it) is very well described, well condensed and is easy to understand. However visual supporting material could be expanded.

Regarding the differences/similarities between species, the differences are well review and are easy to advert. However, the similarities sometimes may be a little confusing to follow. The table is useful for this.

  • In C. elegans part: No mention anything about Plk-2 being activated by HIM-8/ZIM?
  • In C. elegans authors do not mention that telomeres do not form a bouquet. But they do mention it in drosophila. It would be good to mention it in both? (This info is in the table, nonetheless).
  • Table 1- Many lines are not aligned properly (e.g. D. melanogster and M. musculus), it should be at single page it will be easier to follow.
  • Figure 1 has details which are hardly visible without magnification.
  • L 1-16 references missing
  • Line 217: be specific, ‘like in other models’. References missing in the paragraph L210-226
  • Line 252: It is necessary to mention that Spo11 is responsible for the DSB.
  • Line 264: ‘produce fertilized eggs’ is not correct. Say ‘viable’ or ‘fertilizable’ eggs or ‘produce/results in zygotes’.
  • Line 266-268: references missing
  • Line 316: ‘when data are available’ not clear sentence.
  • Line 362: There is an unnecessary ‘ , ‘ after TERB1,
  • L51: No title for yeast

Author Response

In C. elegans part: No mention anything about Plk-2 being activated by HIM-8/ZIM?

We cannot say that PLK-2 is activated per se by the PC-binding proteins (or we missed the info in the literature). However, it is thought that the recruitment of PLK-2 to the nuclear envelope by the PC-proteins allows for the meiosis-specific phopshoprylation by PLK-2 of the Serine 12 of SUN-1, which seems to promote the activity of the LINC complex. We thus have rewritten this part (lines 128-152) to try to be more meaningfull to the reader.

In C. elegans authors do not mention that telomeres do not form a bouquet. But they do mention it in drosophila. It would be good to mention it in both? (This info is in the table, nonetheless).

Indeed. This missing piece of information was added to the text lines 134-152. L139

Table 1- Many lines are not aligned properly (e.g. D. melanogster and M. musculus), it should be at single page it will be easier to follow.

This has been corrected, and we hope it will stay as we formatted it during the journal editing process.

Figure 1 has details which are hardly visible without magnification.

We have tried to increase the size of the small elements.

L 1-16 references missing

Reference 78 has been added, which reviews several articles demonstrating this point. We apologize for this oversight

78.           Alleva, B.; Smolikove, S. Moving and stopping: Regulation of chromosome movement to promote meiotic chromosome pairing and synapsis. Nucleus 2017, 8, 613-624, doi:10.1080/19491034.2017.1358329

Line 217: be specific, ‘like in other models’. References missing in the paragraph L210-226

Now line 228 reads "Like C. elegans, cytoskeletal forces are transmitted to the nuclear envelope and chromosomes by the LINC complex,..."

In line 210-226, we described our work published and reference our article as reference 41

Line 252: It is necessary to mention that Spo11 is responsible for the DSB.

we now mention it directly in the introduction paragraph, which reads "Meiotic DSBs are induced by the topoisomerase-like Spo11, which is conserved in all species studied so far. "

Line 264: ‘produce fertilized eggs’ is not correct. Say ‘viable’ or ‘fertilizable’ eggs or ‘produce/results in zygotes’.

It has been changed to "viable eggs"

Line 266-268: references missing

The following references have been added: "Studies on fixed samples set the stage for further analysis of chromosome movements and dynamics [100,104]. Microtubule inhibitors are known to disrupt both the formation of the bouquet and of the Balbiani body (a structure containing mitochondria, ER and Golgi vesicles) [49]." 

Line 316: ‘when data are available’ not clear sentence.

Line 326 now reads "Most published studies thus analyzed spermatogenesis, and here, we will compare these results with chromosome dynamics in oocytes when data are available in females. "

Line 362: There is an unnecessary ‘ , ‘ after TERB1,

removed

L51: No title for yeast

This special issue of reviews is about "The manufacturing of the oocyte". Yeasts do not produce oocyte and should not be included. However, we felt that studies done in yeasts are so influential for chromosomes dynamics and pairing, that we decided to include them in the introduction as useful reference points for the other organisms. Though, we could not dedicate full parts on yeast .

Round 2

Reviewer 2 Report

No more comments is added